# Function and Plasticity of Electrical Synapses in the Mammalian Brain: Role of Non-Junctional Mechanisms

**DOI:** 10.3390/biology11010081

**Published:** 2022-01-05

**Authors:** Sebastian Curti, Federico Davoine, Antonella Dapino

**Affiliations:** 1Laboratorio de Neurofisiología Celular, Departamento de Fisiología, Facultad de Medicina, Universidad de la República, Montevideo 11800, Uruguay; antonelladapino@fmed.edu.uy; 2Instituto de Ingeniería Eléctrica, Facultad de Ingeniería, Universidad de la República, Montevideo 11300, Uruguay; fdavoine@fing.edu.uy

**Keywords:** gap junctions, connexins, connexons, electrical coupling, Cx36

## Abstract

**Simple Summary:**

Relevant brain functions, such as perception, organization of behavior, and cognitive processes, are the outcome of information processing by neural circuits. Within these circuits, communication between neurons mainly relies on two modalities of synaptic transmission: chemical and electrical. Moreover, changes in the strength of these connections, aka synaptic plasticity, are believed to underlie processes of learning and memory, and its dysfunction has been suggested to underlie a variety of neurological disorders. While the relevance of chemical transmission and its plastic changes are known in great detail, analogous mechanisms and functional impact of their electrical counterparts were only recently acknowledged. In this article, we review the basic physical principles behind electrical transmission between neurons, the plethora of functional operations supported by this modality of neuron-to-neuron communication, as well as the basic principles of plasticity at these synapses.

**Abstract:**

Electrical transmission between neurons is largely mediated by gap junctions. These junctions allow the direct flow of electric current between neurons, and in mammals, they are mostly composed of the protein connexin36. Circuits of electrically coupled neurons are widespread in these animals. Plus, experimental and theoretical evidence supports the notion that, beyond synchronicity, these circuits are able to perform sophisticated operations such as lateral excitation and inhibition, noise reduction, as well as the ability to selectively respond upon coincident excitatory inputs. Although once considered stereotyped and unmodifiable, we now know that electrical synapses are subject to modulation and, by reconfiguring neural circuits, these modulations can alter relevant operations. The strength of electrical synapses depends on the gap junction resistance, as well as on its functional interaction with the electrophysiological properties of coupled neurons. In particular, voltage and ligand gated channels of the non-synaptic membrane critically determine the efficacy of transmission at these contacts. Consistently, modulatory actions on these channels have been shown to represent relevant mechanisms of plasticity of electrical synaptic transmission. Here, we review recent evidence on the regulation of electrical synapses of mammals, the underlying molecular mechanisms, and the possible ways in which they affect circuit function.

## 1. Introduction

Relevant brain functions, such as sensory perception, organization of motor outputs, and cognitive processes, are the outcome of complex functional operations carried out by neural circuits. Within these circuits, communication between neurons mainly relies on two modalities of synaptic transmission: chemical and electrical. Chemical transmission involves the release of neurotransmitter molecules and their binding to specific receptors, which typically results in ionic permeability changes. By contrast, electrical transmission results from the direct spread of ionic current from one neuron to another coupled partner, by means of intercellular ionic channels, typically organized in clusters known as gap junctions.

## 2. Gap Junctions as the Structural Substrate of Electrical Synapses

Electrical synapses are supported by gap junctions, which consist of aggregates of intercellular channels (plaques). In vertebrates, these channels result from the assembly of two hemichannels termed connexons, one from each of the participating cells. In turn, connexons are composed of six subunits (connexins), organized as a hexameric structure around a central aqueous pore. Connexins (Cx) are membrane proteins with four transmembrane domains and three loops (two extracellular and one intracellular), whereas the carboxyl and amino termini are oriented towards the intracellular side. Connexins from apposed cells interact through their extracellular loops [1,2]. These proteins are encoded by 21 different genes in the human and 20 in the mouse genomes, and they are widely expressed in mammalian tissues [3], being designated according to their predicted molecular weight in kilodaltons [4]. Whereas the extracellular loops and transmembrane domains display a certain degree of homology across connexin types, the cytoplasmic loop and C-terminal tail domains show the greatest level of sequence divergence, most probably resulting in differences in terms of permeability, regulation by pH and calcium, and sensitivity to the difference between the membrane potentials of the coupled cells [4]. Gap junctions can assemble, involving identical (homotypic configuration) or different (heterotypic) hemichannels, and each hemichannel can be constituted by identical (homomeric) or different (heteromeric) subunits. Even though there are a hypothetically large variety of possible junctions, as a result of such combinations, most of them are not functional [3,5]. In particular, Cx36, the main synaptic connexin in mammals (see below), appears to be able to form only homotypic gap junctions [6,7]. Moreover, recent evidence suggests that proper function and plasticity of gap junctions depend on the interaction with multimolecular complexes composed of proteins involved in cell adhesion, scaffolding, trafficking, and protein kinases [8,9,10]. For instance, the scaffold protein zonula-occludens-1 has been shown to interact with the C-terminal tail of Cx34.1, homologous to mammalian Cx36 in fish, and its presence is essential for the structure and function of electrical synaptic transmission [11,12]. The cytoskeleton protein tubulin has also been shown to interact with Cx36 at the C-terminal domain, suggesting the existence of a tubulin-dependent transport involved in the regulation of gap junction resistance and size [13]. Furthermore, interaction of the Ca^2+^/calmodulin-dependent kinase II with the cytoplasmic loop and C-terminal domain has been suggested to represent a critical step for the phosphorylation of the kinase’s target site at Cx36 [14,15]. Interestingly, this kinase has been involved in mechanisms of plasticity at electrical synapses supported by Cx36 of the mammalian brain, although the target of these actions has not been identified yet [16,17].

### Intercellular Channels

Gap junction’s channels are permeable to ions as well as small molecules, thus supporting not only the electrical communication between cells but also biochemical signaling through second messengers, such as Ca^2+^, cAMP, and inositol 1,4,5- trisphosphate [18,19,20,21,22,23,24]. As other membrane channels, they present voltage gating processes, as their conductance is sensitive to the voltage difference between cells (transjunctional voltage, *V_J_*), and to a lesser extent, to the membrane potential (*V_m_*) [5,25]. Typically, the *V_J_* dependent gating processes present two mechanisms: “fast” and “slow” (also called “loop”) [26]. The fast voltage gating displays a time constant in the sub-millisecond range, making it nearly instantaneous [27,28,29,30,31]. On the other hand, the slower gating mechanism presents a time constant in the order of seconds and is characterized by the reduction in the gap junction conductance, from its maximal value at *V_J_* = 0, upon changes of *V_J_* on either direction. However, this last mechanism does not produce a total closure of intercellular channels, rendering a fraction of the junctional conductance that is voltage independent. The amount of this fraction from the total conductance, as well as the voltage sensitivity of this mechanism, is highly heterogeneous and depends on Cx composition of intercellular channels [2,27,32,33,34,35,36,37,38,39].

Whereas homotypic gap junctions display a symmetrical dependency of conductance on *V_J_*, heterotypic ones might show asymmetrical relationships that yield diode-like electrical transmission, a property referred to as rectification [40,41,42,43,44,45]. This property, characteristic of some electrical synapses of invertebrates and lower vertebrates, most probably results from disparate gating properties of apposed hemichannels at heterotypic junctions [45,46,47]. Moreover, it has been postulated that asymmetrical conditions of apposed connexons at homotypic gap junctions, such as phosphorylation or cytosolic Mg^2+^ concentration, can also generate rectification of electrical synaptic transmission [45,48].

Despite these gating mechanisms, the low sensitivity of Cx36 channels to *V_J_*, as well as their insensitivity to *V_m_*, suggests that junctional conductance remains stable during substantial fluctuations in membrane potential of connected neurons. In fact, macroscopic conductance of Cx36-based junctions is almost independent of *V_J_* in the region ±40 mV, and then falls to 30–40% of its maximum value, with a time course in the order of seconds [6,35,49]. These characteristics, in combination with the fact that Cx36 only form homotypic (non-rectifying) intercellular contacts, support the idea that, in the context of neural circuits operations, dominated by rapidly varying membrane voltage changes such as action potentials, and low amplitude signals such as synaptic potentials, electrical contacts of mammals behave as voltage-independent junctions that can be modeled as ohmic resistors (see below).

## 3. Distribution of Electrical Synapses

Expression of neuronal connexins is developmentally regulated in a region-specific fashion. For example, Cx43 has been reported in olfactory receptor neurons [50], Cx57 in the olfactory bulb [51] and the retina [52], and Cx45 is expressed in several brain areas such as the neocortex [53] and the retina [54,55]. In spite of that, mammalian neurons predominantly express Cx36, which is considered the main synaptic connexin in juveniles and adults [56,57], as it has been found in the neocortex [58], retina [59], thalamic reticular nucleus (TRN) [60], inferior olive nucleus (IO) [61], mesencephalic trigeminal nucleus (MesV) [62], hippocampus [63], and cerebellum [64], among other structures [65].

Gap junction mediated coupling is prevalent during the embryonic and early postnatal developmental stages in many areas of the mammalian brain [66,67,68], reaching a peak of expression in the first two weeks after birth and decreasing afterwards [69,70]. Whereas these intercellular contacts might aid the maturation of the chemical synaptic circuitry [71,72], the postnatal increase in chemical synaptic activity—associated with the onset of several behaviors—seems to promote the loss of gap junctional connections between developing neurons [73]. For instance, in the hypothalamus, the developmental expression of Cx36-mediated coupling is precisely regulated by chemical synapses. It remains at low levels during early postnatal development, while γ-aminobutyric acid type-A (GABA_A_) receptors have an excitatory effect (as they promote the activation of voltage-dependent Ca^2+^ channels). However, as actions mediated by GABA_A_ receptors turn inhibitory along with metabotropic glutamate II receptors’ activation, Cx36 expression is enhanced, reaching the expression peak between the first and second postnatal week. Afterwards, maturation of the neurotransmission, mediated by N-methyl-D-aspartate (NMDA) receptors, supports calcium entry to neurons and subsequent activation of the cAMP response element binding protein. The activation of this signaling pathway results in a downregulation of Cx36 expression and a reduction in coupling [74,75]. Despite this, electrical neurotransmission persists in the adult brain, connecting fast-spiking (FS) interneurons of the neocortex [76], pyramidal cells of the hippocampus [77,78], primary sensory afferents of the MesV nucleus [62,79], principal cells of the IO [80,81], as well as cone photoreceptor cells [82], ganglion cells [83], and AII amacrine cells [59,84] of the retina, where it plays critical functional roles.

## 4. Electrical Synaptic Transmission

Gap junctions establish pathways of low resistance for the flow of ionic currents, supporting the continuous and fast communication between neurons. In fact, this uninterrupted communication characteristic of electrical synapses sets them apart from most chemical synapses, whose transmission is not usually sustained in time, but it reflects the arrival of action potentials to the presynaptic neuron (although few exceptions have been reported [85,86]). Moreover, presynaptic action potentials at chemical synapses may even fail to trigger neurotransmitter release, for example, in the case of contacts with low release probability or, by the contrary, after vesicle depletion (characteristic of high release probability contacts upon repetitive presynaptic activation). Thus, in contrast to synaptic chemical transmission, which is stochastic in nature, electrical transmission behaves in a deterministic fashion, resulting in a much more reliable modality of interneuronal communication [57,87,88]. Besides, electrical contacts can transmit both depolarizing and hyperpolarizing signals between neurons, unlike chemically mediated synaptic transmission.

According to Ohm’s law, the current through gap junctions (*I_J_*) is proportional to the difference of *V_m_* between the connected cells, or *V_J_*, and inversely proportional to the junctional resistance (*R_J_*). Thus, *V_J_* represents the driving force for the junctional current at electrical contacts. Therefore, a change on the *V_m_* of any of the coupled neurons will induce an instantaneous change in *I_J_* and a corresponding change in the *V_m_* of the other cell, thus supporting electrical coupling between neurons. Since this modality of synaptic transmission relies on the direct flow of current between neurons, it is instantaneous in nature (although not the change in *V_m_*; see below), contrasting with the characteristic delay of chemical synaptic transmission, due to their more complex chain of events.

Transmission at electrical contacts is typically bidirectional in contrast to chemical contacts, which are characterized by the unidirectional flow of information from the pre- to the postsynaptic element, determined by the asymmetrical distribution of the molecular machinery involved. The expression of homotypic non-rectifying gap junction channels is a common trait of electrical contacts in mammals, where direction of current flow is solely dictated by the voltage difference between cells, being from the most depolarized to the less depolarized one. In addition, although heterologous coupling has been reported in the mammalian brain in structures such as the retina [82,89], the dorsal cochlear nucleus [90] or some cortical areas [91], the great majority of electrical coupling so far reported in these animals is homologous, that is, between neurons of the same type [92]. By connecting neurons of similar electrophysiological properties, particularly input resistance, junctional current will have a similar impact on the membrane potential of either coupled cell (see below). Thus, non-rectifying gap junctions, interconnecting neurons of similar electrophysiological properties, result in coupling strength of comparable magnitude in both directions, supporting bidirectional communication at the electrical synapses of mammals. The aforementioned functional differences between electrical and chemical synapses suggest that, instead of mutually exclusive, they represent complementary modalities of synaptic communication. Consistently, available experimental evidence shows that both types of transmission coexist in most regions of the mammalian brain [93,94,95].

### 4.1. Coupling Coefficient

The easiest way to understand the factors that determine the efficacy of electrical synaptic transmission is by analyzing the equivalent circuit representing two neurons coupled by a gap junction, as illustrated in Figure 1A. The intercellular contact is represented by *R_J_*, and each neuron by its membrane capacitance and membrane resistance (*C_m_* and *R_m_*, respectively), whereas active voltage-dependent mechanisms are omitted for the sake of simplicity. As previously mentioned, given that most gap junctions between mammalian neurons are homotypic Cx36/Cx36, these junctions can be modeled as a simple ohmic resistor and current through junctions determined by the Ohm’s law:(1)$$I_{J}=\frac{V_{J}}{R_{J}}$$

A change in the membrane potential of neuron 1 (Δ*V*_*m*1_, presynaptic), establishing a *V_J_* ≠ 0, induces a change in the junctional current *I_J_*, which in turn, produces a voltage deflection in neuron 2 (Δ*V*_*m*2_, postsynaptic), thus electrical coupling. Once the membrane capacitance of coupled neurons is fully charged and current is only resistive (time indicated by vertical double arrows in Figure 1B), the ratio between these two voltage deflections is defined as the coupling coefficient (*CC*) in steady state:(2)$$CC_{SS}=\frac{\Delta V_{m2}}{\Delta V_{m1}}$$

*CC_SS_* measures the steady state efficacy of electrical synaptic transmission, i.e., the attenuation of steady or time unvarying signals (also referred to as DC signals) when transmitted through electrical contacts. For example, a value of *CC_SS_* = 0.1, typical of mammals, means that a sustained presynaptic voltage deflection of 100 mV generates a postsynaptic response of 10 mV.

According to the electrical circuit of Figure 1A, the injection of a current pulse in Cell 1 induces a voltage change in the same cell as well as in the coupled one (Cell 2) (Figure 1B). From these voltage signals, the *CC_SS_* can be calculated, as stated above, and expressed in terms of the circuit resistances:(3)$$CC_{SS}=\frac{\Delta V_{m2}}{\Delta V_{m1}}=\frac{R_{m2}}{R_{m2}+R_{J}}=\frac{1}{1+\frac{R_{J}}{R_{m2}}}$$

A straightforward consequence of this result is that *CC_SS_* is determined only by the ratio between the junctional resistance and the membrane resistance of the postsynaptic neuron, *R_J_* and *R*_*m*2_, respectively [87,96]. So, when *R_J_* >> *R*_*m*2_, the neurons are weakly coupled (*CC_SS_* ≈ 0), whereas if *R_J_* << *R*_*m*2_, the two cells are strongly coupled (*CC_SS_* ≈ 1).

Experimentally, the *CC_SS_* can be measured by injecting long (200–600 ms) current pulses to one neuron and, simultaneously, recording the *V_m_* of the coupled neurons. Reported mean *CC_SS_* at mammalian electrical synapses are quite low: 0.03 to 0.04 in the TRN [60], IO [61,97] and hippocampus [98], 0.09 and 0.13 in FS and low-threshold spiking neocortical neurons, respectively [99]. Nonetheless, it must be noted that gap junctions in these neurons are located at the dendrites, most probably resulting in an underestimation of the *CC_SS_*, as the amplitude of dendritic signals gets considerably attenuated when they reach the neuron’s soma, where the signals are recorded. Consistently, the *CC_SS_*, estimated in pairs of coupled MesV neurons (which are interconnected through somatic gap junctions), almost double those of FS and low-threshold spiking neurons [62].

### 4.2. Frequency Dependence of the Coupling Coefficient

As mentioned before, the *CC_SS_* is a widespread metric used to determine the strength of electrical synapses. However, it may not be representative of the efficacy of transmission of physiologically relevant membrane voltage signals, such as action potentials and chemically mediated postsynaptic potentials, as they are typically faster than the membrane time constant of the postsynaptic neuron. Instead, in order to thoroughly characterize the electrical synaptic transmission, it is more appropriate to characterize the dependence on the frequency (*f*) of the coupling coefficient, *CC*(*f*). According to the circuit in Figure 1A, the presynaptic signal is filtered by *R_J_*, connected in series to the parallel array of the *C*_*m*2_ and *R*_*m*2_. For DC signals, *C*_*m*2_ behaves as an open circuit, and *CC* (0 Hz) = *CC_SS_*, as expected. In contrast, for signals with high frequency content, *C*_*m*2_ will act as a short-circuit that produces Δ*V*_*m*2_ = 0, regardless of the amplitude of Δ*V*_*m*1_. As a result, *CC*(*f*) → 0 as *f* approaches infinity: i.e., the postsynaptic neuron cannot follow the fluctuations of the presynaptic one if they are infinitely fast. These asymptotic results are valid for any electrical synapse, but they do not provide information on the filtering behavior for intermediate frequencies.

The complete frequency-dependence of *CC*(*f*) can be experimentally assessed by injecting, either single- or variable-frequency (also called “zaps”) low-amplitude current sinusoids [100] in the presynaptic neuron, and computing the *CC*(*f*) as the ratio between the Fast Fourier Transforms of post- and presynaptic membrane voltage signals [62,87,101]. Theoretically, the *CC*(*f*) can be expressed by a generalized version of Equation (3), where the input resistance of the postsynaptic neuron is substituted by its complex impedance. If all the intrinsic components were passive, the expected *CC*(*f*) should behave as a low-pass filter. Indeed, that is the case for most reported *CC*(*f*) in the mammalian brain, such as in the neocortex [99], cerebellum [102], TRN [60], and the retina [84,103,104]. However, a more complex behavior in dendrites should not be discarded, where most gap junctions are located, as they express a rich variety of voltage-gated ion channels [105]. Strikingly, a band-pass behavior was reported in the MesV nucleus, due to active ionic conductances operating near the resting membrane potential (see below) [62,87,106].

### 4.3. Spike Transmission

Presynaptic action potentials evoke coupling potentials at the postsynaptic neuron, known as spikelets or postjunctional potentials (PJP) (Figure 1C). These signals are characterized by their smaller amplitude, delayed time to peak, and broader time span, in comparison to the presynaptic spike [40,57,107,108]. Even though the junctional current reacts instantaneously to changes in *V_J_*, the postsynaptic membrane response is delayed by the phase lag introduced by the low-pass filter [62,87,106]. The latency for spike transmission is usually below 1 ms, both in vitro [102,109,110,111] and in vivo [112]. Nevertheless, sub-millisecond latency alone cannot be used as a criterion to identify electrical synapses in the mammalian brain, given that chemical synaptic communication may also be very fast [57,113].

Like the *CC_SS_*, the coupling coefficient for spikes (*CC_spike_*) is defined as the ratio of the PJP amplitude over the presynaptic spike amplitude. However, as fast electrical signals transmitted through junctions are more attenuated than slower ones due to the low-pass filter properties, the *CC_spike_* is typically much smaller than the *CC_SS_*. Accordingly, the *CC_spike_* between neocortical inhibitory coupled interneurons, is about 10 times smaller than its *CC_SS_* [99]. In contrast, spike transmission between MesV neurons is considerably more efficient, given that the *CC_spike_* is only 5 times smaller than the *CC_SS_*, as a result of the band-pass filter properties (see below) [62,114].

The time course of the PJP is not only shaped by the filter properties of electrical synaptic transmission but also by the waveform of the presynaptic action potential. Presynaptic spikes displaying shallow afterhyperpolarization potentials (AHP) elicit monophasic PJP, with a net depolarizing effect on the postsynaptic coupled cell [60,62,84,99,103]. Instead, spikes, followed by deep and long-lasting AHP, produce biphasic PJP [99,102], whose net impact could be more complex (Figure 1C). For example, cerebellar Golgi neurons are electrically coupled and produce very fast action potentials (0.35 ms in half width), followed by a profound and long-lasting AHP (20–30 ms in duration, [102]). The low-pass filtering properties of transmission between these neurons determines that the presynaptic spike gets severely attenuated (*CC_spike_* = 0.007), whereas the AHP is less affected due to its lower-frequency content (*CC_AHP_* = 0.1). Hence, the resulting PJP have a very small and short depolarizing phase corresponding to the presynaptic spike, followed by a stronger hyperpolarizing one corresponding to the AHP, presenting a net inhibitory effect on the postsynaptic neuron excitability [64,102]. Moreover, in FS interneurons of the neocortex, the waveform of the PJP is state-dependent: from depolarized resting membrane potential (RMP) presynaptic spikes are followed by an AHP, resulting in biphasic coupling potentials, whereas from hyperpolarized RMP, spikes lack AHP, and their corresponding PJP are monophasic [115]. In summary, the efficacy of spike transmission through intercellular contacts not only depends on the junctional resistance and electrophysiological properties of the postsynaptic neuron, as discussed before, but also on the shape of the presynaptic spike. As a matter of fact, mammalian neurons express a rich repertoire of voltage-dependent membrane channels, implying that action potential waveform varies considerably among different types of neurons [116]. In that sense, the expression of repolarizing K^+^ currents, is of special relevance, as they are primary determinants of spike duration and, hence, of its frequency content. Therefore, as a general rule, the larger the overlap between the frequency content of the presynaptic spike and the *CC*(*f*) function, the more efficient is the spike transmission [114].

### 4.4. Interaction with the Intrinsic Neuronal Properties

As mentioned before, the electrophysiological properties of the postsynaptic neuron, particularly the membrane resistance and time constant, represent a critical determinant of the efficacy of electrical transmission. Active voltage-dependent conductances, especially those operating near the RMP, confer complex voltage and time-dependent dynamics to *R_m_*, thus endowing parallel characteristics to the *CC*. In fact, beyond the gating processes of intercellular channels, active electrophysiological properties of coupled neurons endow coupling potentials with fast and highly sensitive voltage dependency. For instance, the persistent Na^+^ current (I_NaP_) is an inward current that is swiftly activated by depolarizations near the RMP and lacks inactivation. Its activation results in an increase in the apparent *R_m_*, thus acting as a boosting mechanism of subthreshold depolarizations [117,118]. Consistently, in the context of synaptic electrical transmission, the I_NaP_ operates as an amplifying mechanism for depolarizing PJP, increasing the coupling coefficient in a voltage-dependent manner, as was shown in fish [119] and mammals [62,102]. Moreover, due to its lack of inactivation, the involvement of the I_NaP_ not only increases the PJP amplitude but also its duration, further accentuating its excitatory effect. Interestingly, in coupled TRN neurons, this mechanism acts in a state-dependent way, only when they are in the tonic spiking regime, but not when they are in bursting mode. This results from the difference in RMP at which each behavior is expressed: tonic spiking occurs when the RMP is within the activation range of I_NaP_ (around −53 mV, close to the I_NaP_ half-activation voltage), while bursting happens at more hyperpolarized RMP (around −76 mV) [120].

Moreover, K^+^ currents, activated by subthreshold depolarizations, such as the A-type current (I_A_), might also impart relevant properties to electrical synaptic transmission. Nevertheless, in contrast to the I_NaP_, I_A_ activation repolarizes the *V_m_*, and due to its slower kinetics, it temporally curtails depolarizing potentials slower than its activation time constant. In this way, the I_A_ selectively dampens slow depolarizations, whereas fast changes of the *V_m_* are unopposed, thus acting as a high-pass filter mechanism [101]. Consistently, I_A_ is responsible for shaping the band-pass filter behavior of the *CC*(*f*) in MesV neurons, by introducing a resonant bump around 50–80 Hz, due to its time constant [62]. In these neurons, I_A_ diminishes the gain of *CC*(*f*) for signals slower than its activation time constant, being counterbalanced by I_NaP_-induced boosting. These mechanisms also reduce the phase lag of the transmission. Thus, acting together, the I_NaP_ and the I_A_ increase the efficacy of electrical transmission in a frequency dependent fashion, promoting the synchronic activation of pairs of coupled MesV neurons [62,114]. This emphasizes the notion that voltage operated ion channels of the non-junctional membrane might significantly contribute to the properties of electrical transmission between neurons.

## 5. Functional Role of Electrical Synapses in Neuronal Circuits

Electrical synapses endow coupled neurons with a simple, yet sophisticated, tool for information processing. In spite of that, the functional contribution of electrical synapses to neuronal circuits has been underestimated historically, in comparison to the chemical synaptic communication [88,121]. As a matter of fact, electrical and chemical synaptic transmission coexist in several neuronal circuits, acting together to perform complex operations. Beyond synchronization, electrical synapses have been recognized as critical contributors to complex operations performed by ensembles of coupled neurons. Here, we will succinctly review some outstanding examples of circuit operations that rely heavily on electrical synapses. For an up-to-date compilation of neuronal circuits where electrical synapses play a major role, see [122].

### 5.1. Synchronic Firing

Early descriptions already recognized the synchronizing capability of electrical synapses [107,123]. In fact, the fast and bidirectional communication through gap junctions provides an effective mechanism to share excitation among electrically coupled neurons, thus promoting the synchronic activation of coupled neurons [57,108]. Indeed, this temporally correlated activity heavily relies on the transmission of the slow pre-potential that precede spikes, which is facilitated by the low-pass filter properties. Besides, a spike in a leading neuron will induce a spikelet in the follower one, adding extra depolarization to the latter and accelerating the trajectory of its membrane potential to the firing level, thus promoting that both cells reach threshold at the same time [112]. Supporting this idea, modeling studies [124,125,126,127], as well as dynamic clamp experiments, in which artificial electrical synapses connect otherwise uncoupled neurons [128], clearly demonstrates the role of electrical coupling in synchronization. Moreover, as these coherent activity patterns depend on the mutual excitatory influences among clusters of coupled neurons, synchronization tends to be more robust between neurons of the same type (homologous coupling). In fact, electrical coupling between neurons of similar electrophysiological properties, interconnected by non-rectifying gap junction channels, typically results in widespread in-phase activity within these networks [92,127]. Consistently, millisecond-scale synchronization between electrically coupled neurons has been shown in the neocortex [99,129], the TRN [60], the olfactory bulb [130,131], the striatum [132], the retina [133], the suprachiasmatic nucleus [134], the cerebellar cortex [64,102,112,135], the IO [97], the MesV [62], as well as between sympathetic preganglionic neurons [136], among other structures.

Nonetheless, electrical synapses might also support desynchronization of neuronal activity [64,132]. For example, electrically coupled cerebellar Golgi cells present spontaneous synchronic activation that, upon sparse mossy fiber input, results in spike desynchronization. This inhibitory effect is caused by propagation of the spike AHP through gap junctions. As mentioned, spikes in Golgi cells are characterized by a fast-depolarizing phase followed by a long lasting AHP. The low-pass filtering properties at these contacts determine that coupling potentials are predominantly hyperpolarizing [102]. When mossy fiber inputs activate only one neuron of a network of heterogeneously coupled cells, the AHP propagation has variable inhibitory effect on surrounding cells according to their coupling strengths. This triggers network desynchronization as it introduces spike-phase dispersion [64]. However, in vivo recordings in anesthetized mice showed that the spikelet has a predominantly excitatory effect, with its afterhyperpolarization appearing to be less functionally relevant than under in vitro conditions. This difference may arise from the voltage-dependent boosting of the depolarizing component of the PJP, promoted by the more depolarized membrane potentials observed in vivo [112]. Thus, according to this study, the role of electrical coupling in the network of cerebellar Golgi cells is to ensure millisecond-scale spike synchronization, both in the context of spontaneous activity and during sensory-evoked synaptic drive [112].

### 5.2. Lateral Excitation

As previously mentioned, electrical coupling provides a mechanism to share excitation within neural circuits, hence creating functional compartments for signal pooling. Such mechanisms might support lateral excitation by which the activity in some cells helps to activate neighboring inactive ones, thus, operating as a positive feedback mechanism. Although it might degrade synaptic specificity within neural circuits, lateral excitation between primary afferents tuned to qualitatively similar stimuli, operates as a positive feedback, amplifying sensory responses critical for the organization of motor outputs as suggested in invertebrates and lower vertebrates [137,138,139,140]. In the mammalian brain, this mechanism has also been postulated to enhance excitability in networks of electrically coupled neurons of the olfactory bulb [141], the cerebellar cortex [142] and the dorsal cochlear nucleus [90,143], where excitability of inactive neurons is enhanced by the spread of current from active coupled neurons.

The functional relevance of electrical synapses is perhaps best exemplified in the vertebrate retina, where the ubiquity of Cx expression strongly suggests they are critically involved in processing of visual information [144,145,146]. Amongst these functional operations, lag normalization in ON-OFF directionally selective ganglion cells (DSGC) constitutes an amazing example. There are four subtypes of DSGC, each of which responds preferentially to edges moving in a specific cardinal direction. The spatial location of the edge is signaled by the initiation of spiking activity in DSGC, whereas their peak firing rates encode the stimulus velocity [147]. Interestingly, only a subpopulation of DSGC are electrically interconnected through their dendrites [148], via homotypic Cx45-containing gap junctions [104,149,150]. Whereas the dendritic arbors of uncoupled DSGC are symmetrical, in coupled cells they are aligned with their preferred direction [151]. Characteristically, in uncoupled cells, spiking only starts when the leading edge of a light bar is inside their dendritic tree, and the timing depends on its velocity. By contrast, in coupled DSGC, firing begins when the incoming edge is 100 μm of their somas and dendrites, irrespective of the input velocity, in a phenomenon designated “lag normalization” [152]. This effect is mediated by electrical coupling, as stimulated neurons send anticipatory signals to yet unstimulated neurons, and these priming signals allow early activated DSGC to recruit their coupled neighbors even before the leading edge arrives [152]. Thus, electrical synapses between DSGC provide a lateral excitation mechanism, enabling a precise detection of the spatial location of the stimulus, disregarding its velocity [153].

### 5.3. Oscillatory Activity

Rhythmicity is a key aspect of biological systems and has particular relevance in brain functions such as motor pattern generation. Such rhythmic patterns might rely on the activity of automatic pacemaker neurons or, alternatively, on an emergent property of a network of interconnected neurons [154,155,156]. The relevance of electrical coupling in the generation of rhythmic activity by neural circuits has been studied by means of theoretical approaches, suggesting that the network architecture and number of coupled neurons, as well as the strength of coupling and electrophysiological properties of participating neurons, are all key factors determining spike frequency [157,158,159]. However, regardless of the compelling experimental evidence about electrical coupling in the mammalian brain, its role in generating rhythmic activity is not so clearly established. For instance, neurons of the IO forms one of the most extensive electrically coupled neuronal networks in these animals [160,161], and characteristically display synchronous spontaneous subthreshold oscillations (STO), both in vitro [162,163] and in vivo [164]. These neurons present electrical resonant properties that enable them to act as oscillators, mainly due to the presence of T-type Ca^2+^ and H-type currents [165,166]. Nevertheless, it has not been completely clear if the STO derived exclusively from the intrinsic resonant characteristics of single IO neurons, or from an emergent network property, as the dissection of the contribution of these two components is not straightforward. Pharmacological experiments using the gap junction blocker glycyrrhetinic acid, showed that, while the synchronization across the population of IO neurons is completely lost, as expected, STO frequency and amplitude are normal, suggesting that electrical coupling is not critical for STO generation [167]. Nonetheless, concomitant changes in neuronal properties due to off target effects of this blocker, does not allow it to rule out the involvement of electrical coupling. Supporting these results, it has been shown that cells in both wild-type and Cx36 knock-out mice generate STO of similar frequency and amplitude. However, these oscillations in pairs of neighboring wild-type neurons are strongly synchronized, whereas in Cx36 knock-out pairs they are temporally uncorrelated. This reinforces the idea that IO neurons are single-cell oscillators synced by electrical coupling, and that gap junctions between olivary neurons are not essential for generation of oscillations in the wild-type animal [61]. However, uncoupled IO neurons from the mutant animal present unusual voltage-dependent oscillations and increased excitability. These changes, attributable to a combined decrease in leak conductance and an increase in voltage-dependent calcium conductance, suggest that neurons compensate for the lost network connectivity, by tuning their intrinsic electrophysiological properties, in order to maintain the capability to produce rhythmic activity [168]. Further evidence in this line was obtained, employing a Cx36 knock-down model generated with a lentivirus-based vector, to block gap junctional coupling. This study revealed that robust and continuous subthreshold oscillations require gap junctional coupling of IO neurons, supporting the notion that network oscillations represent an emergent property of a population of electrically coupled weak and episodic single-cell oscillators [169], consistent with a previous theoretical study [158]. According to that, experimental and theoretical evidence, indicating a central role of electrical coupling in supporting rhythmic activity, was also obtained in cerebellar Golgi cells [102] as well as in neuroendocrine tuberoinfundibular dopamine neurons of the hypothalamus [170].

### 5.4. Coincidence Detection

Electrical synapses may also endow circuits of coupled neurons with the ability to selectively respond to excitatory synchronic inputs [62,84,109]. In fact, when a neuron receives an input, part of the underlying current flows through gap junctions towards coupled cells. Thus, electrical synapses act as current sinks, reducing the input resistance of all cells of the network (Figure 2A). Since coupled cells act as a “load”, this effect is known as “loading”, and it results in a great reduction in the excitability of coupled neurons [171]. This leak of current through junctions might represent a considerable fraction of the neuron’s input resistance [57,62], and the extent of the loading effect is positively correlated with the coupling strength [172]. In contrast, synchronic depolarizing inputs to all coupled cells induce parallel variations of their membrane potentials, reducing the voltage drop across junctions and minimizing the flow of current to coupled cells. In this way, simultaneous inputs mitigate the loading effect (Figure 2B). Under this condition, also known as “unloading”, changes in membrane potential of all neurons of the network are of bigger amplitude, facilitating their activation (Figure 2C). This property allows electrical coupling to maximize the impact of coincident inputs while dampening temporally dispersed ones, supporting coincidence detection (Figure 3) [172,173,174,175,176,177]. In the retina, such a mechanism has been proposed to play a relevant functional role, as it underlies a noise reduction operation. In fact, electrical coupling between cone photoreceptors and between AII amacrine cells determines that simultaneous (signal specific) inputs have a larger impact on membrane potential, in comparison to randomly distributed (noisy) ones, increasing the signal-to-noise ratio [144,178,179].

Modeling results of a large-scale network of coupled neocortical FS neurons showed that, despite synchronization, the main consequence of the presence of gap junctions is a reduction in excitability. In fact, electrical coupling reduces the total number of spikes generated in response to temporally dispersed synaptic inputs in the network. This reduction in spike firing is due to shunting through the gap junctions (loading condition). In contrast, synchronization of inputs during a temporal window, as narrow as 20 ms, is enough to induce a dramatic firing increase in the network, showing that the population of coupled FS interneurons may function, collectively, as detectors to synchronized synaptic inputs in the millisecond-scale range [175]. These results show that the output of circuits of coupled neurons is critically determined by the temporal dispersion of its inputs, reinforcing the idea that they might operate as coincidence detectors (Figure 3). This property has been experimentally explored by alternatively injecting synchronous or asynchronous near threshold depolarizing current pulses to pairs of electrically coupled neurons. This approach applied to FS neocortical neurons [180], AII amacrine cells [84], MesV neurons [62] and cerebellar basket cells [181], showed a significant increase in firing or spike probability during synchronous pulses in comparison to temporally dispersed ones (see below). Thus, coincident inputs to circuits of coupled neurons are signaled by burst firing, which are considered reliable codes of information between neurons [182]. Since neuronal spiking represents the functional expression of coincidence detection, a difference in firing during simultaneous versus uncorrelated inputs is considered a direct indicator of this circuit function [183]. Thus, the increase in firing during simultaneous activation, in comparison to when cells are independently activated, represents the gain of coincidence detection or the susceptibility of networks of coupled neurons to coincident inputs [172]. Interestingly, in the MesV nucleus, it has been shown that, while electrical coupling is absolutely necessary for coincidence detection to occur, it is not possible to establish a direct correlation between coincidence detection gain and coupling coefficient. This indicates that the gain of coincidence detection cannot be explained solely in terms of the coupling strength. Indeed, high susceptibility to coincident inputs was displayed, almost exclusively, by strongly coupled pairs, while the reciprocal does not occur, indicating that strong coupling is a necessary condition, although not sufficient, for high gain coincidence detection. Moreover, highly excitable neurons tend to be more susceptible to coincident inputs, unlike less excitable ones. This strongly suggests that the intrinsic excitability of neurons critically contributes to set the gain of this relevant functional operation in networks of coupled neurons, which was confirmed by modeling results [172].

## 6. Plasticity of Electrical Synaptic Transmission Supported by Non-Junctional Mechanisms

Changes in the efficacy of transmission at electrical synapses might redirect the flow of information within neural networks with relevant functional consequences. By reconfiguring neural circuits, such phenomenon of synaptic plasticity endows animals with the ability to adapt to changes imposed by fluctuating environmental and physiological conditions, as was shown in the vertebrate retina [145,184], or even support learning, as was shown in central-pattern generating networks of Aplysia [185]. As mentioned before, the coupling strength depends on the junctional resistance and on the membrane resistance of the postsynaptic cell, as indicated in Equation (3). Rearranging this equation, the *CC_SS_* can easily be expressed as:(4)CCSS=Rm2RJ1+Rm2RJ

By defining a dimensionless variable x=Rm2RJ, the *CC_SS_* can be analyzed as a monotonically increasing function of only one variable (x), thus:(5)$$CC_{SS}=\frac{x}{1+x}$$

Non-conductive gap junctions, corresponding to *R_J_*
→∞, yields x≈0 and *CC_SS_*
≈0, resulting in uncoupled cells as expected. On the other hand, as *R_J_*
→0 and/or *R*_*m*2_
→∞, x takes larger values, and *CC_SS_*
≈1. The *CC_SS_* as a function of x is plotted in Figure 4A according to Equation (5). Despite this theoretical analysis, it is interesting to note that average values of *CC_SS_* in electrical synapses of mammals typically lie below 0.3 (red trace in Figure 4A,B), implying that small changes in x, result in large modifications of the *CC_SS_*. Therefore, from this analysis it is clear that, on the one hand, changes of either the junctional resistance or the membrane resistance of the postsynaptic cell equally impact on the variable x, which, in turn, result in modifications of the coupling strength. On the other hand, most synapses exist in the region where subtle modifications of either junctional or non-junctional mechanisms (*R_J_* or *R_m_* respectively), have a significant impact on the coupling. This suggests that, regardless of the mechanism, plasticity is a constitutive property of electrical synaptic transmission.

While plasticity of electrical synaptic transmission due to modulation of the junctional resistance is widely acknowledged [94,106,184,187], the emerging role of the electrophysiological properties of coupled neurons is much less recognized. Therefore, we next discuss how the regulation of ion channels of the non-junctional membrane could represent a powerful mechanism of synaptic plasticity, which adds further complexity to electrical transmission through neuronal gap junctions.

### 6.1. Modulation by Ligand-Gated Channels of Chemical Synapses

Early work in the IO already suggested modulation of electrical coupling by regulation of electrical properties of the non-junctional membrane [80]. IO neurons are electrically coupled via dendritic gap junctions located at special structures known as glomeruli, which represent unique microcircuits formed by the convergence of 4–10 dendritic spines from different neurons [188]. Each glomerulus is also innervated by both GABAergic and glutamatergic inputs [189]. In particular, the inhibitory GABAergic input was proposed to play an important role in dynamically regulating the strength of coupling between IO neurons through a shunting effect, thus acting as an uncoupling mechanism [80]. Therefore, by activating GABA_A_ receptors, this inhibitory input decreases the membrane resistance in close proximity to the electrical synapses, reducing the coupling strength between neurons, according to Equations (4) and (5). The final demonstration of the shunting hypothesis was provided 40 years later by transfecting neurons from the deep cerebellar nuclei with channelrhodopsin-2 [81]. Photoactivation of GABAergic terminals evokes inhibitory postsynaptic potentials at IO neurons and a considerable reduction in the coupling coefficient, revealing the modulation of electrotonic coupling by chemical neurotransmission. Moreover, this work also showed that the directionality of electrical transmission can be regulated by this inhibitory GABAergic input, thus modifying the functional architecture of the IO network [81]. Interestingly, asynchronous GABA release was demonstrated at contacts between neurons of deep cerebellar nuclei and IO neurons, which provides powerful and sustained inhibitory actions, strongly suggesting the effective regulation of electrical coupling between IO neurons [190]. Likewise, the uncoupling of IO neurons by GABAergic cerebellar inputs abolishes subthreshold oscillations, supporting the idea that oscillations are, in part, an emergent network property that depends on electrical coupling (see above) [81,191].

### 6.2. Modulation by Voltage-Gated Channels of the Non-Junctional Membrane

Besides neurotransmitter-gated channels, modulation of voltage-gated channels of the non-junctional membrane has also been implicated in plasticity of electrical synapses and the functional operations they support. In the MesV nucleus, the upregulation of HCN channels, which support the hyperpolarization-activated cationic current (I_H_), results in less intuitive effects on coincidence detection [172]. I_H_ is exceptional among voltage-gated membrane conductances, as it presents a unique ion selectivity and gating properties. In fact, it is a mixed cationic current carried mainly by Na^+^ and K^+^, determining a reversal potential of about −30 mV in physiological conditions. Additionally, unlike most membrane conductances, it is activated by hyperpolarizing voltage changes rather than depolarizations. Thus, upon hyperpolarizations from resting potential, I_H_ activation results in an inward current that depolarizes the membrane potential with a slow time course. Moreover, activation of I_H_ is facilitated by cyclic nucleotides (cAMP and cGMP) by way of a direct interaction with an intracellular domain [192]. This modulatory action consists of an acceleration of its gating kinetics, as well as a displacement of its activation curve towards more positive membrane potentials, determining activation even at RMP. In spite of modulation, extreme slow activation kinetics determines that, in the time course of single spikes, I_H_ mainly operates as a leak current with a depolarized reversal potential. Thus, activation near the resting potential contributes to setting both the membrane voltage and neuronal membrane resistance. Intriguingly, despite this apparent simplicity, neuromodulatory actions on this current may have unexpected effects on electrical excitability of neurons. Indeed, upregulation of this current results in an increase in the net inward current that promotes firing by bringing membrane voltage closer to its firing level. On the other hand, its activation also results in a reduction in the neuron’s input resistance dampening the impact of excitatory input. Which of these outcomes predominate on excitability seems to be related to the neuronal type and cellular compartment in which HCN channels are expressed [193,194,195,196].

Interestingly, experimental and computational work on coupled MesV neurons showed that the upregulation of I_H_ preferentially increases firing during coincident depolarizations, as spiking during independent depolarizations showed little changes (Figure 5A,B) [172]. This suggests that during independent stimulation, the increase in net inward current and the reduction in *R_m_* due to HCN upregulation compensate for each other. Instead, during coincident inputs, the loading by coupled neurons is cancelled, mitigating part of the *R_m_* reduction. Under this condition, the net inward current increase prevails over the reduction in the *R_m_*, boosting neuronal excitability and supporting strong repetitive discharges. Moreover, by lowering the *R_m_* of coupled cells, upregulation of HCN channels reduces the coupling strength according to Equations (4) and (5). Despite that, the upregulation of I_H_ increases the contrast of neuronal output in response to uncorrelated versus coincident inputs. Therefore, by enhancing the susceptibility of pairs of coupled MesV neurons to coincident inputs, upregulation of HCN channels increases the gain of coincidence detection, despite a modest reduction in the coupling strength (Figure 5B) [172].

Interestingly, the I_H_ and electrical coupling have been demonstrated in IO neurons [97,197], GABAergic neurons of TRN [60,198], pyramidal neurons from the hippocampus [78,199] and the prefrontal cortex [200,201], Golgi cells of the cerebellar cortex [102,202], bipolar cells of the retina [103,203], and mitral cells of the olfactory bulb [130,204], among others. This shows that the expression of HCN channels is a common trait in electrically coupled neuronal populations, strongly suggesting that these two mechanisms constitute a functional motif of the mammalian brain. This poses interesting perspectives about the regulatory control on functional operations supported by electrical coupling such as coincidence detection, as the I_H_ current has been established as a highly modifiable membrane mechanism.

## 7. Concluding Remarks

Electrical synapses represent a widespread form of interneuronal communication of the mammalian brain. Besides the apparent simplicity of their underlying mechanism, the interaction with the electrophysiological properties of neurons results in complex emergent properties with relevant functional operations. Theoretical analysis supports the notion that most electrical contacts are prone to plastic modifications, as small changes of either the junctional resistance or the electrophysiological properties of coupled neurons tend to be translated into significant modifications of the efficacy of transmission. Despite modulatory actions on gap junction channels, ion channels of the non-junctional membrane are in an exceptional position for mediating such phenomenon of synaptic plasticity. In fact, control of the membrane resistance, by way of nearby chemically mediated synaptic inputs, represent a strong and fast mechanism to modify the strength of electrical coupling. Moreover, voltage-gated channel function is also highly modifiable through the action of neuromodulators, deeply impacting the electrophysiological properties of neurons and, therefore, on the transmission through electrical contacts. It is tempting to speculate that persistent changes on the efficacy of electrical transmission could eventually contribute to processes of memory, learning, and physiological regulations.

## Figures and Tables

**Figure 1 biology-11-00081-f001:**
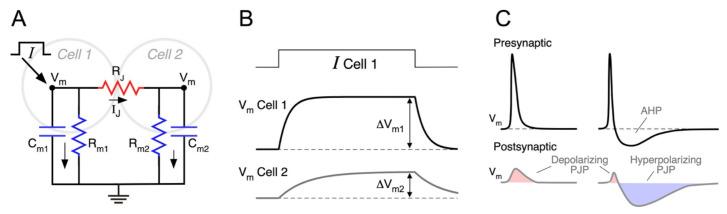
Properties of electrical coupling between neurons. (**A**) Equivalent circuit of a pair of electrically coupled cells, including the membrane resistance of Cell 1 (*R*_*m*1_) and Cell 2 (*R*_*m*2_), their corresponding membrane capacitances (*C*_*m*1_ and *C*_*m*2_), as well as the junctional resistance (*R_J_*). Injection of a current pulse in the node corresponding to the intracellular side of Cell 1 (oblique arrow), establishing the flow of current, indicated by the smaller black arrows. (**B**) Injection of a depolarizing current pulse in Cell 1 (*I* Cell 1, top trace) results in a voltage membrane change in the same cell (*V_m_* Cell 1, middle trace) and a corresponding membrane voltage change in the coupled cell (*V_m_* Cell 2, bottom trace). Amplitude of membrane voltage changes in each cell at steady state (once *C*_*m*1_ and *C*_*m*2_ are fully charged) are indicated (Δ*V*_*m*1_ and Δ*V*_*m*2_, vertical double arrows). These values are employed to calculate the coupling coefficient at steady state (*CC_SS_*) (see text). (**C**) Representative illustration of the membrane potential of coupled neurons showing the impact of presynaptic action potential waveform and low-pass filter properties on postjunctional potentials (PJP). Left: presynaptic spikes lacking afterhyperpolarization potentials (AHP) elicit monophasic PJP, with a net depolarizing effect on the postsynaptic coupled cell. Right: instead, spikes followed by deep and long-lasting AHP, produce biphasic PJP with a net hyperpolarizing effect on the postsynaptic coupled neuron.

**Figure 2 biology-11-00081-f002:**
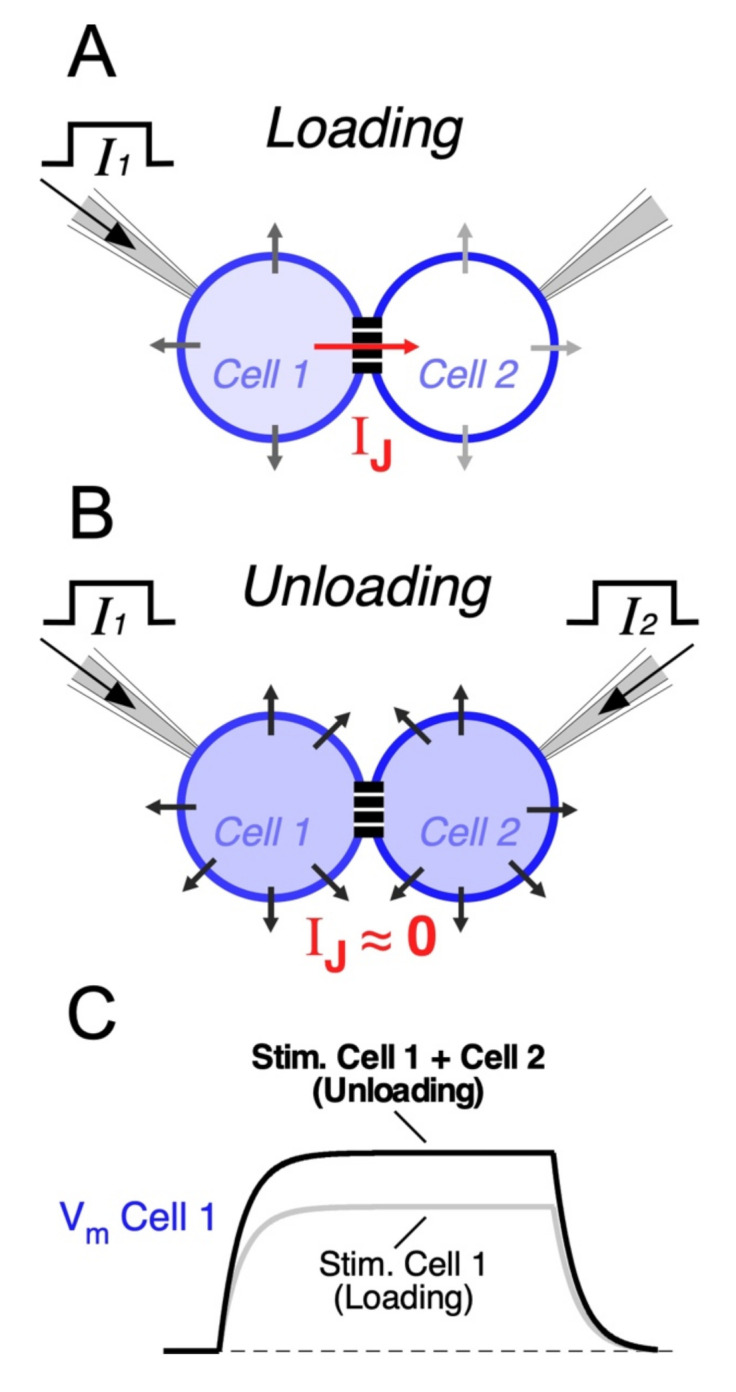
The loading effect in circuits of electrically coupled neurons. (**A**) Injection of a current pulse into one cell belonging to an electrically coupled pair (*I*_1_, Cell 1). While most of the current flows through the membrane of the same cell, a fraction of it will flow through the junction (*I_J_*) and through the membrane of the coupled cell (Cell 2). (**B**) In contrast, simultaneous current injection in both coupled cells (*I*_1_, Cell 1 and *I*_2_, Cell 2) induces similar voltage changes in each cell making *V_J_* close to 0, thus eliminating the flow of current through the gap junction. (**C**) Schematic representation of membrane voltage changes of Cell 1 (*V_m_* Cell 1) in each of the aforementioned conditions. As the leak of current through junctions tends to be eliminated during the simultaneous injection of current into both cells, membrane voltage response in Cell 1 (Unloading, Stim. Cell 1 + Cell 2, black trace) has higher amplitude in comparison to when current pulse is applied only to Cell 1 (Loading, Stim. Cell 1, gray trace). Therefore, the loading by electrically coupled cells results in a reduction in the effective input resistance of Cell 1, as part of the current leaks to Cell 2 through the gap junction.

**Figure 3 biology-11-00081-f003:**
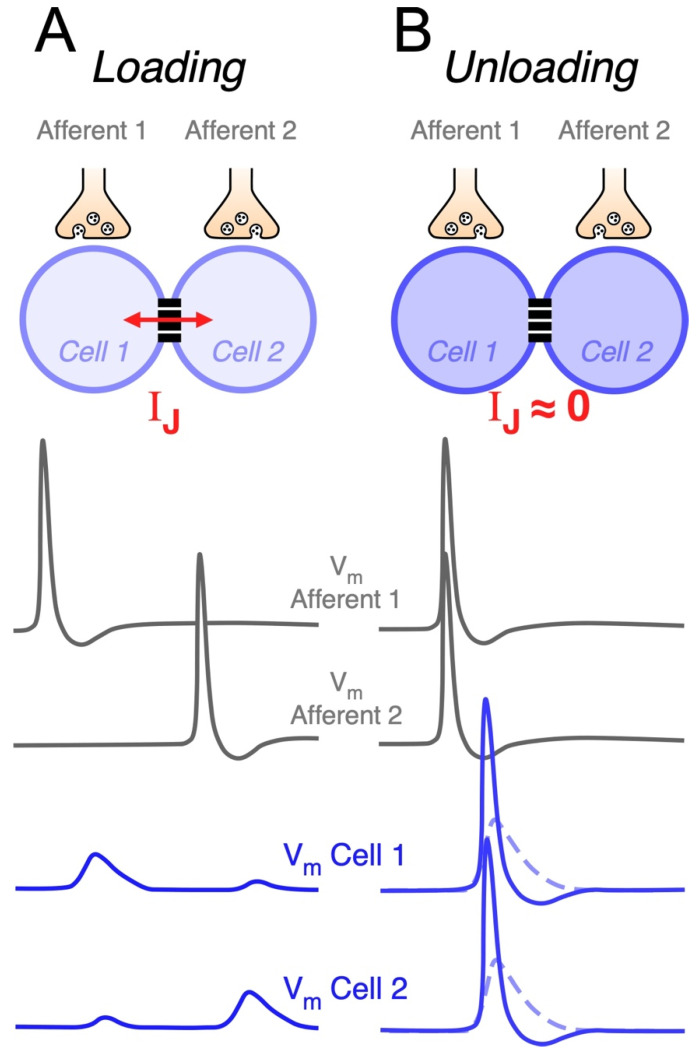
Electrical synapses enable neural circuits to perform coincidence detection. Above, schematic representation of a neural circuit composed of a pair of coupled neurons (Cell 1 and Cell 2). Each neuron receives an independent chemically-mediated excitatory synaptic input by way of corresponding afferents (Afferent 1 and Afferent 2). Traces represent schematic illustration of membrane potential of presynaptic afferents (grey, middle panel) and of membrane potential of postsynaptic electrically coupled neurons (blue, lower panel). (**A**) Asynchronous activation of Afferent 1 and Afferent 2 results in postsynaptic potentials in Cell 1 and Cell 2, respectively, and corresponding nearly instantaneous coupling potentials in the neighbor cell due to the spread of current through the junction (*I_J_*) (Loading). (**B**) Coincident activation of Afferent 1 and Afferent 2 results in simultaneous postsynaptic potentials at coupled neurons (dashed traces), minimizing the voltage drop across the junction and hence, the junctional current (*I_J_* ≈ 0). Cancelation of the loading effect evokes postsynaptic potentials of larger amplitude that facilitate the activation of coupled neurons (continuous traces) (Unloading).

**Figure 4 biology-11-00081-f004:**
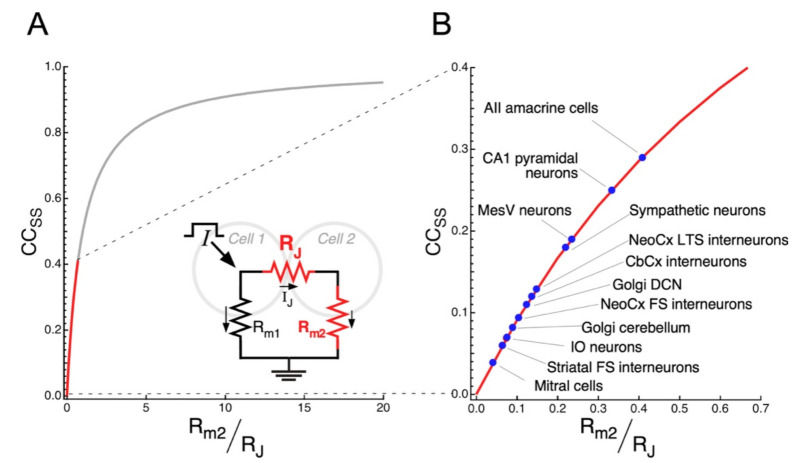
Determinants of electrical coupling and typical values in the mammalian brain. (**A**) Plot of the *CC_SS_* as a function of the ratio of the membrane resistance of the postsynaptic cell (*R*_*m*2_) to the junctional resistance (*R_J_*), according to Equation (5) in main text. Red trace depicts the region of the curve, corresponding for typical values of *CC_SS_* in electrical synapses of mammals. Drawing shows the equivalent circuit for a pair of coupled cells during current injection into Cell 1 (oblique arrow, *I*). As the *CC_SS_* is calculated from membrane voltage changes at steady state, when membrane capacitance is fully charged and current is only resistive, the circuit includes only the membrane resistance of coupled cells (*R*_*m*1_ and *R*_*m*2_) and *R_J_*. Smaller arrows indicate the direction of current flow in the circuit. (**B**) Portion of the plot in *A* illustrated at larger scales, indicating the average values of *CC_SS_* reported for mitral cells [131], striatal fast spiking (FS) interneurons [132], inferior olive (IO) neurons [97], cerebellar Golgi cells [64], neocortical fast spiking (FS) interneurons [99], Golgi cells of the dorsal cochlear nucleus (DCN) [186], interneurons of the cerebellar cortex molecular layer [135], neocortical low-threshold spiking interneurons [99], sympathetic interneurons [136], mesencephalic trigeminal (MesV) neurons [62], CA1 pyramidal neurons [78], and AII amacrine cells of the retina [84].

**Figure 5 biology-11-00081-f005:**
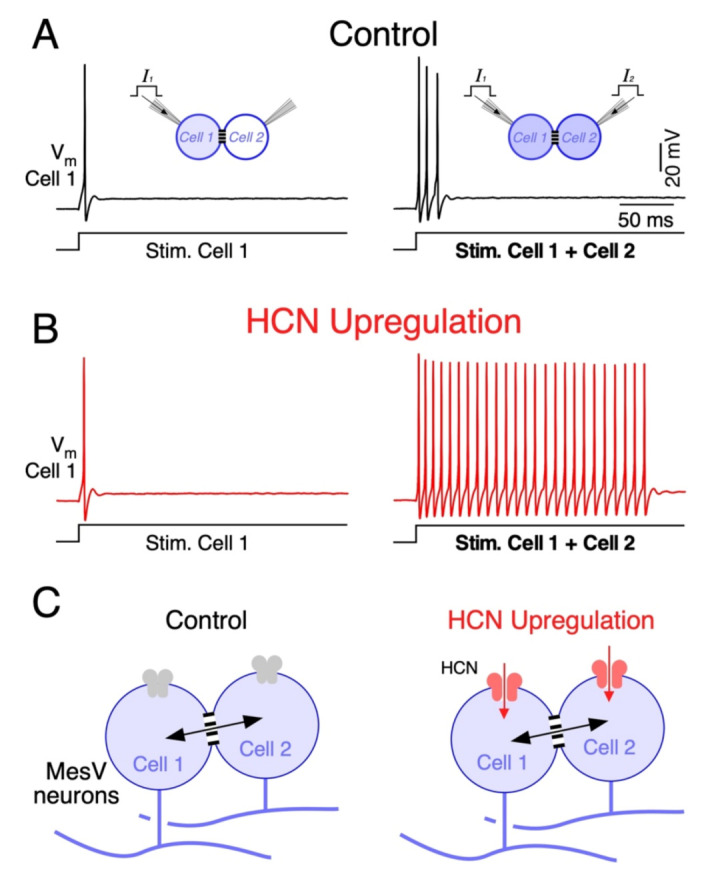
Upregulation of HCN channels enhances coincidence detection by circuits of electrically coupled MesV neurons. (**A**) Membrane voltage responses of one MesV neuron, belonging to an electrically coupled pair (*V_m_* Cell 1), to the injection of depolarizing current pulses of the same intensity (300 pA). Left: Cell 1 activation (Stim. Cell 1); right: simultaneous activation of both coupled neurons (Stim. Cell 1 + Cell 2) to show coincidence detection in control conditions. (**B**) Membrane voltage responses of the neuron depicted in *A* to the same stimulation protocol after upregulation of the I_H_ by application of cGMP (1 mM). This result shows a dramatic increase in firing when coupled cells were simultaneously activated, indicating an increase in coincidence detection gain. Traces in *A* and *B* represent whole cell patch clamp recordings in current clamp, obtained by standard methods [172]. (**C**) Schematic representation of a pair of coupled MesV neurons in control conditions (left) and after upregulation of HCN channels, responsible of the I_H_ current (right). Upregulation of these channels support an inward current that, in combination with the unloading effect, results in a selective increase in firing during synchronic depolarizing inputs.

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
