# Peer review of "Function and Plasticity of Electrical Synapses in the Mammalian Brain: Role of Non-Junctional Mechanisms"

_biology, 2022, doi:10.3390/biology11010081_

Round 1

Reviewer 1 Report

THis is a thorough, comprehensive, well-illustrated and well written review of the impact of voltage dependent nonjunctional conductance on electrical coupling through gap junction channels.

My specific comments are intended to clarify certain sections where meaning may not be obvious.

l18.  Any conductance change caused by ligand will also affect such transmission, and this should be mentioned because the impact of voltage dependent conductances may be more esoteric to most readers.

l75.  For most connexins minimum unitary condutance gammajmin is about 10% gammajmax

l162.  THis sentence is garbled

l181 change on to in

l189 typical of mammalian electrical synapses (CC in cardiac myocytes is about 1)

1206.  THis is a very important point and should be very clear.  THe next sentence contains phrases "as the CC50...." that I think may confuse the reader and I suggest deleted the phrases

l224.  Also transfer of metabolically relevant fuel and metabolites

l278  This could be illustrated.

l292  What does this mean?

l355  What does use to be more robust mean here?

l367.  PSPs would have similar effects

l436.  Rephrase for clarity

l491 and Fig 3.  THe meaning of the lower blue panels and the red arrows within them is unclear.  In addition to potential changes due to transmitter, there also are conductance changes

l535 and fig 4 and Eqns 4,5..  Use of gj here instead of Rj in Fig 1A is confusing, and use of dot rather than x to indicate the product is also not so clear.  I suggest using R, and changing access to Rm2/Rj.

Reviewer 2 Report

This is an excellent review that will be a nice addition to the literature. I only have a few minor comments:

  1. There is an excessive number of un-necessary abbreviations. For example, there is no need to abbreviate connexin in the abstract. Some abbreviations are defined and then never used, like thalamic reticular nucleus (TRN) , inferior olive nucleus (IO) [49], mesencephalic trigeminal nucleus (MesV).
  2. Figures 4 and 5 appear to contain summaries of previously published data or new results. They need appropriate citations or detailed methods.

Reviewer 3 Report

In their review "Function and plasticity of electrical synapses in the mammalian brain: role of non-junctional mechanisms" the authors offer a comprehensive analysis of electrical signaling in neuronal transmission and the development of plasticity circuits. The importance of gap junctions, specifically those comprised of Cx36 are discussed and the biophysical (electric) properties of these channels are summarized. 

Overall the review is very well written, positing important notions that will likely promote further investigation into the contributions of electrical synapses in neural circuitry. The figures are appropriate and designed well, and help to clarify some of the complex physics concepts regarding the electrophysiology, which to this reviewer are valuable as this is not my area of expertise beyond interpretation of trace data.  

I have only one suggestion: In the introductory paragraph of Section 2, the authors discuss regulators of gap junctions and the importance the divergence of the CL and CT in differential regulation. I think it is important to note that these domains (the CT especially) are also major hubs of protein-protein interactions which contribute heavily to the regulation and transitions in stages of the Cx life cycle (See doi: 10.3390/ijms19051428 for a review).  It would be interesting to expand the review to include known synaptic protein interactions that modulate the electrical properties of synaptic gap junctions and the resultant electrical synaptic conduction (if studies exist). 

Reviewer 4 Report

The authors present a detailed review and argument for the  impact of gap junction-mediated modulation of neural circuits in coupled neurons in the brain.  These arguments are best developed from sections 5 forward, with only minor word choice or syntax errors.  Sections 1-4, however, are less well-written and would require reconfiguring and re-writing to brign them up to the level of the rest of the manuscript.

  1. The authors should make clear that the discussion in the manuscript pertains to coupled neurons in mature mammalian brain.  While Cx36 is primarily a mature neuronal connexin, other Cx are expressed in neurons during development.  The behavior and electrophysiology of glial connexins would be a separate category for discussion.
  2.  Careful use of synaptic vs gap junction/junctional currents should be checked, specifically in section 4.
  3. While the use of acronymns can be a necessary shorthand,  the author's might consider using fewer to enhance readability for a more general audience.
  4. The paper, while generally well-written, contains a number of awkward phraisings and word choices. 
  5. It should be noted that the more technical arguments were laid out clearly and fully.
